# Adolescent Water Safety Behaviors, Skills, Training and Their Association with Risk-Taking Behaviors and Risk and Protective Factors

**DOI:** 10.3390/children7120301

**Published:** 2020-12-17

**Authors:** Isabell Sakamoto, Sarah Stempski, Vijay Srinivasan, Tien Le, Elizabeth Bennett, Linda Quan

**Affiliations:** 1Seattle Children’s Hospital, Seattle, WA 98105, USA; sarah.stempski@seattlechildrens.org (S.S.); tizbenharb1@gmail.com (E.B.); linda.quan@seattlechildrens.org (L.Q.); 2School of Public Health, University of Washington, Seattle, WA 98195, USA; vs6563@uw.edu; 3School of Science, Technology, Engineering and Mathematics, University of Washington Bothell, Bothell, WA 98011, USA; tienle98@uw.edu; 4Harborview Injury Prevention Center, University of Washington School of Medicine, Seattle, WA 98195, USA

**Keywords:** injury prevention, drowning, water safety, adolescent, life jacket, swimming lessons, swimming ability, risk behaviors, protective factors

## Abstract

*Background*: Drowning remains the third leading cause of unintentional injury death for adolescents in the United States. *Aims:* This study described adolescent swimming lessons, behaviors (life jacket wear while boating) and comfort (swimming in deep water) and their association with protective and risk factors and risk-taking behaviors reported by Washington State students in Grades 8, 10, 12, primarily comprised of youth ages 13 to 18 years. *Methods*: This study used the 2014 Washington State Healthy Youth Survey (HYS), a publicly available dataset. *Results*: Most students reported having had swimming lessons, using life jackets, and comfort in deep water. Differences reflected racial, ethnic and socioeconomic disparities: being White or Caucasian, speaking English at home and higher maternal education. Lowest rates of comfort in deep water were among Hispanics or Latino/Latinas followed by Blacks or African Americans. Greater life jacket wear while boating was reported by females, those in lower grades and negatively associated with alcohol consumption, sexual activity and texting while driving. Having had swimming lessons was associated with fewer risk-taking behaviors. *Conclusions*: The HYS was useful to benchmark and identify factors associated with drowning risk among adolescents. It suggests a need to reframe approaches to promote water safety to adolescents and their families. Multivariate analysis of this data could identify the key determinants amongst the racial, ethnic, and socioeconomic disparities noted and provide stronger estimation of risk-taking and protective behaviors.

## 1. Introduction

Drowning is the third leading cause of injury death among adolescents 15 to 19 years in the United States (U.S.) [1]. From 2014 to 2018, the U.S. drowning death rate of adolescents ages 15 to 19 (1.1 per 100,000) was twice that of those ages 10 to 14 (0.5 per 100,000) [1]. Males were 5.3 times more likely to drown than females and Black or African American youth were 2.6 times more likely to drown compared to White or Caucasian youth [1]. In Washington State during this time period, the drowning death rate among those 15 to 19 (1.5) eclipsed other pediatric age groups [1]. Drownings in this age group in Washington State primarily involve swimming, playing, and boating in open water, like lakes or rivers [2]. 

High drowning rates among teens may be explained by their developmental stage. Adolescents are susceptible to peer pressure and engagement in risk-taking behaviors with a greater focus on rewards rather than costs and consequences [3]. In surveys and focus groups, adolescents have reported behaviors that put them at greater risk for drowning [4,5]. Risk factors such as neighborhood disorder may contribute to youth engaging in risk behaviors [6]. Fortunately, family and peer protective factors have proved effective in reducing risk behaviors like alcohol consumption at an early age [4]. It’s unclear whether these behaviors are associated with other risk behaviors. Moreover, no studies have evaluated protective factors among adolescents with regards to drowning risk. To examine the risk and protective factors and risk-taking behaviors for drowning amongst Washington State adolescents, we evaluated self-reported risk and protective factors and risk-taking behaviors among adolescents taking the Washington Healthy Youth Survey (HYS). The HYS is an adapted version of the U.S. nationally administered Youth Risk Behavior Surveillance System (YRBSS), which is administered by the Centers for Disease Control and Prevention (CDC). Both the YRBSS and HYS include six categories of health-related behaviors that contribute to the leading causes of death and disability among youth and adults [7]: Behaviors that contribute to unintentional injuries and violenceSexual behaviors related to unintended pregnancy and sexually transmitted diseases, including HIV infectionAlcohol and other drug useTobacco useUnhealthy dietary behaviorsInadequate physical activity

We conducted a preliminary study to examine risk and protective associations between risk and protective factors and risk-taking behaviors with life jacket wear and formal swimming lesson participation amongst Washington State adolescents. Our goal was to use the HYS to better understand drowning risk and protection in the context of other adolescent risk-taking behaviors. Understanding those relationships could better identify which drowning factors might be associated with other general risk-taking and protective factors and better inform water safety promotion and drowning prevention interventions. Our objective was primarily to estimate the proportion of Washington youths who have had formal swimming lessons, wore life jackets (also known as personal flotation devices or PFDs) while boating and were comfortable in water over their head and characterize them. Additionally, we sought to evaluate risk-taking and protective factors around adolescents’ wearing life jacket and having had swimming lessons.

## 2. Materials and Methods

### 2.1. Population

The HYS is conducted every two years to students in age Grades 6, 8, 10, or 12, primarily comprising youth ages 13 to 18 years [8]. All Washington public schools, except institutional or correctional schools, are invited to participate in the survey. Private schools can opt-in if they choose. Parents can decline their child’s participation. Student participation is completely voluntary and anonymous. 

Survey data from 2014 were used in analysis of this study. To create a statewide sample for 2014, Washington State schools were randomly selected by representative agencies of HYS administration (Washington State Department of Health; Office of the Superintendent of Public Instruction; Department of Social and Health Services’ Division of Behavioral Health and Recovery; Liquor and Cannabis Board; and the contractor, Looking Glass Analytics, Inc., Olympia, Washington). A total of 35,262 students and 192 schools contributed to the statewide sample. All information obtained for this study is publicly available from www.AskHYS.net. 

We limited this study to students in Grades 8, 10 and 12 surveyed in 2014 because these grades received the Forms with the questions of interest and this was the only year that all drowning prevention questions were included.

### 2.2. HYS Survey

The HYS surveys students for their health concerns, behaviors, school climate, quality of life, mental health, risk and protective factors and risk-taking behaviors. Behavioral questions addressed topics of alcohol, marijuana, tobacco, other drug use, sexual behaviors, and behaviors associated with intentional or unintentional injuries.

In this study, we focused on adolescents’ responses to water safety questions regarding evidence-based drowning prevention interventions that decrease risk for drowning: a history of formal swimming lessons (i.e., lessons that were taught by a swim instructor are or that are received as part of another activity such as day care, school or camp) [9], use of life jackets [10,11] and swimming ability [5]. 

Students in grades 8, 10, and 12 completed Forms A/A-enhanced and B/B-enhanced [12]. Surveys were distributed randomly among students. This study assessed responses from B/B-enhanced only because it included all water safety questions. Form B consisted of 116 items derived primarily from the Youth Risk Behavior Survey and the Global Youth Tobacco Survey and included life jacket and swimming questions [12]. Form B-enhanced included six additional optional questions on sexual orientation, behavior and abuse. We assessed students who responded to the questions regarding swimming lessons (Q5), comfort swimming in deep water (Q6), and life jacket wear while boating (Q7). (Appendix A
Table A1).

The HYS is available in English and Spanish. Spanish speaking students read a translated survey but respond on the English answer sheet to preserve anonymity [12]. Surveys were administered to all participating students in a single class period by a trained test administrator during the school day. Students completed surveys individually. Completed surveys were sealed and returned to Looking Glass Analytics by survey coordinators [12].

### 2.3. Variables

Demographic variables collected included student’s gender, race and ethnicity, maternal education, language spoken at home, living situation, living arrangement due to finances, and grade level (details of these questions are located in Appendix A
Table A1). We selected HYS questions regarding adolescent risk and protective factors and risk-taking behaviors by cross-referencing Jessor’s Domains of Adolescent Risk Behavior and the 2014 HYS Questionnaire Form B/B Enhanced [13]. Categorization of chosen questions reflected the HYS framework for risk and protective factors and risk-taking behaviors [8]. HYS questions were reviewed for validity and reliability by experts in public health, injury prevention, and school health prior to inclusion in the administered survey [8]. The behaviors and factors chosen include unintentional injury behaviors (e.g., life jacket wear, exposure to swimming lessons), protective factors (e.g., eating dinner with family, having supportive adults), and risk-taking behaviors and risk factors (e.g., texting while driving, alcohol use, involvement in physical fighting) (Appendix A
Table A1). These factors were not specifically related to water behavioral factors, but general lifestyle factors. 

The three HYS survey questions related to drowning risks were: (Q5) have you ever taken formal swimming lessons?; (Q6) I am comfortable playing and swimming in water over my head; and (Q7) how often do you wear a life jacket when you’re in a small boat like a canoe, raft, or small motorboat? (Appendix A
Table A1). These questions were developed by drowning prevention experts at Seattle Children’s Hospital, Washington State Department of Health and Public Health Seattle and King County and reviewed and approved by Washington State Department of Health prior to their inclusion in the HYS. Life jacket wear (Q7) was added to the HYS starting in 2002, and both swimming questions (Q5 and Q6) were added in 2014. 

For this study, we used responses to the question about comfort in water (Q6) as a validated indicator of swim ability [5]. Taking formal swimming lessons and wearing life jackets are both validated drowning prevention indicators [9,10].

### 2.4. Analyses

This study used descriptive statistics. Respondents and their responses to each of the three water safety questions were analyzed separately. 

The HYS includes the Q x Q analysis tool, publicly available through www.AskHYS.net [14]. This was used to analyze results for HYS questions by cross-tabulating two variables at a time for comparison. Collapsed versions for question responses were used for analysis by selecting the “collapsed” option in the query builder. Chi-Square tests were conducted using Microsoft Excel to estimate *p*-values to inform statistical significance of risk-taking behaviors, risk and protective factors associated with life jacket use and swimming lessons [15].

This Seattle Children’s Institutional Review Board deemed that IRB review and approval was not required for this study due to its not involving human subjects.

## 3. Results

### 3.1. Descriptive Statistics

A total of 26,163 valid surveys from students in Grades 8, 10, and 12 across 167 schools were received for the 2014 HYS. Non-completion rates of Form B were 17% of Grade 8, 12% of Grade 10, and 9% of Grade 12; and for Form B-enhanced 17% of Grade 8, 14% of Grade 10, and 12% of Grade 12 [12]. A total of 10,456 students in these grades completed Form B or Form B-enhanced and responded to the question: “How often do you wear a life jacket when you’re in a small boat like a canoe, raft, or small motorboat?” (Q7)—Of these, 2631 students indicated that they were never on a small boat; 13,120 students responded to the question: “Have you ever taken formal swimming lessons?” (Q5); and 13,095 students responded to the question: “I am comfortable playing and swimming in water over my head” (Q6). 

The majority of students in Grade 8 were ages 13 to 14, Grade 10 were ages 15 to 16, and Grade 12 were ages 17 to 18 [8]. The majority of respondents identified as White or Caucasian; English was the primary language spoken in the majority of households (Table 1). 

Of the students who answered Q5, 60.4% reported having had formal swimming lessons. Male and female respondents had similar rates (59% versus 61.8%). Those who had had versus not had swimming lessons differed greatly in race and ethnicity, language spoken at home and maternal education. Most students who had had swimming lessons were White or Caucasian (60.7%) while those who had not had swimming lessons were non-White or Caucasian (62%), of which Hispanics or Latino/Latinas were the largest group. By race and ethnicity, White or Caucasian respondents had the highest rate (70.6%) of having had swimming lessons, followed by Asian or Asian Americans, Native Hawaiian or other Pacific Islander respondents (58.9%), and Black or African American respondents (49.4%). The lowest rate of swimming lessons (30.9%) was reported by Hispanic or Latino/Latina respondents. Among those whose mothers had had higher education, 75% of students had swimming lessons compared to 43.9% of students whose mothers had less education (*p* < 0.01). Similarly, among those whose primary language at home was English, 65.8% had had swimming lessons compared to only 26.6% of those whose primary language at home was Spanish. 

Of the students who had been on small boats, 56.7% reported having worn life jackets. Females’ wear rate was higher than males’ (59% versus 53.8%). Notably, respondents who reported wearing life jackets versus not wearing them on small boats differed in language spoken at home and school grade. Those who reported life jacket wear while boating were more likely to speak English at home (56.6% versus 32.1%, *p* < 0.01). In contrast, respondents who did not wear life jackets were more likely to be Hispanic or Latino/Latina (11.7%) and not speak English at home (18.4%) (Table 1). Life jacket wear while boating decreased as grade level increased. Those who reported usually wearing a life jacket were less likely to be comfortable in water (87.7% versus 92.7%, *p* < 0.01).

Impressively, 86.7% of the responding students reported they were comfortable in deep water (Table 1). Respondents who were comfortable versus were not differed in race and ethnicity, maternal education, language spoken at home and grade in school. Those who were comfortable were mostly White or Caucasian (56.6%), spoke English at home (82.2%) and had higher maternal education (58.2%) (Table 1). When evaluated by race and ethnicity, almost all White or Caucasian respondents (92%) reported being comfortable in deep water; Black or African American respondents reported the lowest rates of comfort in deep water (70.3%). The percent comfortable swimming in deep water was highest among 8th graders and decreased with each subsequent grade.

### 3.2. Risk-Taking Behaviors and Risk Factors

Life jacket wear while boating was negatively associated with several risk-taking behaviors examined (Table 2). Students reporting life jacket wear while boating were less likely to have ever consumed alcohol than non-life jacket wearers (87.1% versus 69.9%, *p* < 0.01); less likely to have ever initiated alcohol use (60.3 versus 37.4%, *p* < 0.01) and if they had, were less likely to have initiated alcohol use at <14 years of age (23.4% versus 37.3%). They were also less likely to have ever had sexual intercourse (24.5% versus 32.2% *p* < 0.01) and less likely to be involved in physical fights (17.7% versus 28.9%, *p* < 0.01) compared to non-life jacket wearers. However, life jacket wearers were less likely to have used condoms than non-wearers (13.7% versus 21.7%, *p* < 0.01). Conversely, life jacket non-wear was associated with most risk-taking behaviors and risk factors examined, including alcohol use, sexual activity, involvement in a physical fight, and texting while driving.

Students who had had swimming lessons and those who had not differed in prevalence of several risk-taking behaviors. Respondents who had swimming lessons were less likely to report initiation of alcohol at a young age (if at all), sexual activity, involvement in a physical fight, and texting while driving. 

### 3.3. Protective Factors

Differences in the presence of protective factors were noted between life jacket wearers and non-wearers as well as those with and without a history of swimming lessons (Table 2). 

All protective factors were significantly higher in those who wore life jackets and in those who had swimming lessons. Notably, respondents who reported wearing a life jacket when in a small boat were significantly more likely to usually have had dinner with their family than not (68.1% versus, 57.8%, *p* < 0.01). Similarly, those who had swimming lessons versus not were more likely to usually have had dinner with their family (67.0% versus 54.0%, *p* < 0.01). They were more likely to have adult support (70.5% versus 60.1%, *p* < 0.01), feel safe at school (90.1% versus 84.1%, *p* < 0.01), and think about consequences before making decisions (86.2% versus 81.9%, *p* < 0.01). 

Those who had had swimming lessons were much more likely to be comfortable swimming in deep water relative to those who had not or were unsure (92.6% versus 77.6%, *p* < 0.01). While large percentages of both life jacket wearers and non-wearers were comfortable in deep water, those comfortable in deep water were less likely to wear a life jacket than to wear one (87.7% versus 92.7%, *p* < 0.01). 

## 4. Discussion

This evaluation of a large number of teenage students across Washington State provided estimates of adolescents’ life jacket wear, swimming lessons history and swimming ability. It demonstrated racial, ethnic and socioeconomic disparities and possibly underlying behavioral patterns that may underlie the differences.

The HYS can be useful as a surveillance tool to track life jacket wear, swimming lessons and swimming ability among adolescents—markers that have not previously been widely assessed. The reported 56% life jacket wear correlates well with observational studies of life jacket use among boaters [16]. The self-reported swimming lesson rate of 60% is the first estimate available for this age group but cannot be corroborated since swim programs do not report student enrollment numbers. Importantly, in this study, males and females reported similar rates of formal swimming lessons. Thus, disparities in swimming lessons cannot explain the marked disparity seen in male versus female fatal drowning rates where in the United States, between 2014 and 2018, the male death rate for unintentional drownings was 1.63 while the female death rate was 0.21 [1].

That 87% of students felt comfortable in water over their heads was surprising. Those comfortable swimming in deep water were more likely to have had swimming lessons and less likely to wear a life jacket. We used respondents’ reports of being comfortable in water as a proxy for their swim ability based on a prospective study of school age students’ estimation of their swim ability; being a “good swimmer” or “comfortable in deep water” correlated with the ability to pass a swim test at a public pool [5]. However, that validation study primarily involved school age youth of families with high income levels and few older adolescents. Importantly, surveys of older adolescent New Zealand males suggested that they overestimated their swim abilities [17]. Teens may overestimate their swim skills and abilities and underestimate water hazards—both of which increase their risk for drowning—due to their impulsive tendencies, lack of swimming experience, and peer pressures [18]. Additionally, others have reported that swimming ability is associated with lower life jacket wear [19].

This large cross-sectional study is the first to confirm race-based and socioeconomic differences associated with both swimming lesson history and swim ability among teenage students across a state. White or Caucasian versus non-White or Caucasian race/ethnicities and higher versus lower level of maternal education, the proxy for socioeconomic status, were inversely related to both swim history and swim ability. Previous surveys in the U.S., using multivariate methods, have shown racial and ethnic differences in swim competencies with Black or African American teens reporting lower swim abilities than White or Caucasian teens [20,21]. However, these studies addressed only large urban populations, especially inner-city children. A national survey of New Zealand high schoolers demonstrated racial and ethnic differences in swim competencies [22]. Like this study, it did not identify whether race or income level were independent predictors of swim competency. 

A study of high school students in Turkey identified a significant relationship between adolescent rates of accidental injuries during school and their parents’ education level similar to our findings [23]. The same study suggested the benefit of school and parental supports and policies for preventing injuries in adolescents [23]. In one U.S. study, youths’ swim ability was better if parents could swim and encouraged them to swim [21]. In Thailand, using multiple logistic regression, swim ability was associated with income level, formal swimming lessons and guardian’s swim ability [24]. Thus, families and schools may provide protective factors to prevent drowning. One next step in water safety promotion could be to encourage protective factors, like prosocial involvement and family involvement, to potentially reduce engagement in risk-taking behaviors and thus reduce drowning risk [25]. 

These findings also showed a consistent association between language spoken at home and responses to all three water safety behaviors and skills. Language spoken at home is a proxy for immigrant status, a variable rarely collected. Many countries have reported increased drowning risk among their immigrant populations [26]. In fact, Canada has focused its drowning prevention program on its “new Canadians” which includes a linguistic approach [27,28]. Water safety education programs must include input from culturally diverse and immigrant populations throughout program development and implementation and provide information for parents in languages other than English. Furthermore, it’s essential that water safety programs recruit and retain lifeguards, swimming instructors, program administrators, and educators that reflect the communities that they aim to reach [26,28]. 

The results of this preliminary study support the usefulness of the HYS to explore the relationship of other adolescent behaviors and drowning risks. The HYS provided a unique opportunity to evaluate adolescents’ water safety skills, behaviors and comfort in the context of risk-taking and risk and protective factors. The associations between risk-taking behaviors with life jacket wear and swimming lessons may imply that youth who don’t wear life jackets or have formal swimming lessons are at greater risk for other adverse health-related conditions and events, like drowning. 

We found that teenagers who wore a life jacket or had formal swimming lessons also experienced more positive familial and school relationships compared to those who did not wear a life jacket and did not have swimming lessons. Thinking about consequences before making a decision is one of 20 internal assets within the evidence-based Developmental Assets Framework [29]. One of their social competencies is planning and decision making indicating that a young person knows how to plan ahead and make choices. Drowning prevention recommendations could be included as part of skills training for adolescents related to decision making and risk reduction. Participation in swimming lessons may contribute to behavioral protective factors like critical thinking before making decisions. Programs should consider protective factors in the development of water safety and drowning prevention interventions and messaging as an approach to mitigate known risk factors for drowning [26]. Life jacket use and ability to swim, along with other water safety skills are considered core components of water competency [30]. Considering risk and protective factors and risk-taking behaviors within the context of water competency could be useful. 

This study suggests a need to reframe how water safety is promoted to adolescents. One systematic review of multiple health risk behaviors (MHRBs) amongst adolescents presents the effectiveness of multiple health risk interventions—particularly in schools—for health-risk behaviors [31]. Incorporating drowning prevention as a component of widespread adolescent prevention programs in schools may be ideal for student participation and engagement in water safety education, and reinforcement of positive social norms, like life jacket wear, between peers and teachers [31]. 

Finally, this study identified a role of socioeconomic as well as cultural, language and racial differences in life jacket wear while boating and exposure to swimming lessons among adolescents. Students who wore life jackets and those who had had swimming lessons were mostly White or Caucasian and primarily spoke English at home. Conversely, life jacket non-wear was associated with being male and having low maternal education/lower income. Racial and ethnic differences were most marked between respondents who had swimming lessons versus not. White or Caucasian students were far more likely and Hispanic or Latino/Latina students least likely to have had swimming lessons compared to any other race studied. These differences underscore the need to provide access to swimming lessons among non-White or Caucasian, immigrant and lower income communities. Swimming programs should include organizations that reflect and serve culturally diverse communities to promote inclusivity and help reduce racial, ethnic and socioeconomic disparities in adolescent water safety behaviors and skills [25].

### 4.1. Limitations

This study has five limitations. First, the population studied did not represent all adolescents since it excluded teens who had dropped out of school and may not have included a representative sample of adolescents who attended private schools. Second, students with limited reading proficiency or those who spoke a language other than English or Spanish may not have had the same opportunity to complete the survey. Third, findings may not be generalizable as Washington State has had a strong drowning prevention focus that promoted life jacket use for over twenty years. Fourth, the self-report data may be influenced by memory, social desirability, reading ability, and developmental changes and tend to be overestimated [12]. However, the correlation between reported and observed life jacket use while boating rate was reaffirming [16]. This study was cross-sectional so potential correlations can only be inferred. 

Finally, this study was unable to evaluate the associations using individual based responses which would allow complex univariate or multivariate analysis. The HYS dataset is publicly available but restricted in the amount and type of data available. To perform further analyses including multivariate analyses required access to individuals’ data which required review and approval by the Washington State Department of Health and Institutional Review Board. Due to demands of the COVID-19 pandemic, limited state resources precluded review by both entities in our allotted time frame. 

### 4.2. Future Research

Future studies could explore:How exposure to protective factors influences adolescents’ attitudes and behaviors towards life jacket use and adoption of water competencies such as learning CPR and self-awareness of swimming limitation [30,32].The role of access to swimming lessons versus cultural attitudes in increasing swim comfort amongst racial/ethnic groups [25].Family and community factors in providing youth with swimming lessons and the scale of this impact.The relationship between having had formal swimming lessons and engaging in high-risk behavior.Differences between urban, suburban and rural youth.

## 5. Conclusions

This study demonstrated the usefulness of the Washington State HYS dataset to evaluate drowning risk and protection in the context of other adolescent risk-taking behaviors. The findings provide surveillance data but also new insights into drowning injury. They suggest the need to reframe our thinking and approach to promoting water safety to adolescents and their families. Findings align with previous studies’ that youth experiencing low socioeconomic status and racial and ethnic disparities—specifically Black and African American and Hispanic and Latino or Latina—are at a significant disadvantage concerning swimming ability and water safety behaviors [20,21,33]. This study suggests identifying risk-taking adolescents as being at greater risk for drowning, identifying non-White or Caucasian and low income communities for improved access to swimming lessons, and a more holistic approach to life jacket use in teens by addressing risk-taking behaviors, risk and protective factors and emphasizing that swim ability does not mean that a life jacket is needed less often. Further evaluation of associations between risk-taking behaviors and life jacket usage may help identify psychosocial predecessors of risk-taking behaviors. 

## Figures and Tables

**Table 1 children-07-00301-t001:** Demographic characteristics of 2014 HYS Form B/Form B-enhanced Respondents’ Life Jacket Wear While Boating, Exposure to Swimming Lessons and Comfort in Deep Water **.

Demographics	Life Jacket Wear When Boating		Had Swimming Lessons		Comfortable Swimming in Deep Water	
	Usually	Not Usually	Yes	No/Not Sure	Strongly Agree/Agree	Disagree/Strongly Disagree
Total N and Percentage	*N* = 5924 (56.7%)	*N* = 4532 (43.3%)	*N* = 7930 (60.4%)	*N* = 5190 (39.6%)	*N* = 11,351 (86.7%)	*N* = 1744 (13.3%)
**Gender**						
Male	2817 (47.6)	2423 (53.5)	3825 (48.3)	2658 (51.2)	5719 (50.5)	754 (43.4)
Female	3096 (52.3)	2104 (46.5)	4089 (51.3)	2526 (48.7)	5615 (49.5)	985 (56.6)
**Race/Ethnicity**						
American Indian or Alaskan Native	132 (2.2)	157 (3.5)	182 (2.2)	180 (3.3)	328 (2.9)	34 (2.0)
Asian or Asian American, Native Hawaiian or other Pacific Islander	701 (11.9)	373 (8.3)	876 (10.8)	612 (11.4)	1183 (10.5)	302 (17.5)
Black or African American	199 (3.4)	164 (3.6)	486 (6)	498 (9.3)	399 (3.5)	168 (9.7)
Hispanic or Latino/Latina	559 (9.5)	527 (11.7)	543 (6.7)	1213 (22.6)	1364 (12.1)	385 (22.3)
White or Caucasian	3447 (58.5)	2616 (58.0)	4910 (60.7)	2041 (38.0)	6390 (56.6)	555 (32.1)
More than one selected/other	852 (14.5)	668 (14.8)	1093 (13.5)	822 (15.3)	1623 (14.4)	285 (16.5)
**Mother’s Education**						
High school or less	1366 (32.4)	1262 (29.9)	1591 (21.2)	2031 (40.2)	2961 (27.8)	660 (41.1)
More than high school	3420 (61.2)	2428 (57.5)	5080 (67.6)	1706 (33.8)	6191 (58.2)	589 (36.7)
Don’t know/doesn’t apply	801 (14.3)	532 (12.6)	846 (16.7)	1315 (26.0)	1492 (14.0)	358 (22.3)
**Language spoken at home**						
English	4805 (91.7)	3517 (81.7)	6581 (86.1)	3422 (70.1)	8925 (82.2)	1063 (64.6)
Spanish	312 (6.0)	339 (7.9)	298 (3.9)	823 (16.9)	830 (7.6)	287 (17.4)
Other	124 (2.4)	451 (10.5)	763 (10.0)	636 (13.0)	1103 (10.2)	296 (18.0)
**Live with most of the time**						
Parents/guardians	5337 (94.2)	3888 (90.7)	7181 (67.0)	1250 (71.6)	10,040 (92.9)	1488 (90.3)
Not parents/guardians	330 (5.8)	400 (9.3)	3541 (33.0)	496 (28.4)	764 (7.1)	159 (9.7)
**Current living arrangements the** **result of losing home**						
No/not sure	5394 (95.3)	3991 (93.9)	7209 (95.4)	3516 (91.4)	10,168 (94.9)	1500 (92.3)
Yes	265 (4.7)	260 (6.1)	344 (4.6)	330 (8.6)	547 (5.1)	126 (7.7)
**Grade**						
8	2711 (45.8)	1411 (31.1)	3161 (40.0)	2154 (41.5)	4621 (40.8)	674 (38.6)
10	1916 (32.3)	1686 (37.2)	2646 (33.4)	1816 (35.0)	3842 (33.8)	616 (35.4)
12	1297 (21.9)	1435 (31.7)	2123 (26.8)	1220 (23.5)	2888 (25.4)	454 (26.0)

** Column percentages.

**Table 2 children-07-00301-t002:** Risk-taking behaviors, risk and protective factors associated with life jacket use and swimming lessons **.

	Life Jacket Wearing When Boating		Swimming Lessons	
	Usually	Not Usually	Yes	No/Not Sure
**Risk-Taking Behaviors**				
**Current alcohol consumption on any days *** No days Any days	5135 (87.1) 763 (12.9)	3149 (69.9) 1357 (30.1)	1489 (18.8) 6412 (81.2)	1017 (19.7) 4134 (80.3)
**Number of lifetime sexual partners *^,+^**				
Never	765 (69.9)	495 (54.2)	1683 (74.6)	984 (67.1)
1 to 3 people	279 (24.5)	294 (32.2)	433 (19.2)	361 (24.6)
4 or more	63 (5.5)	124 (13.6)	139 (6.2)	122 (8.3)
**Condom usage during last sex act**				
Yes	236 (13.7)	259 (21.7)	332 (14.7)	287 (19.5)
No	143 (8.3)	195 (16.4)	237 (10.5)	207 (14.1)
Never had sex	1340 (78.0)	739 (61.9)	1684 (74.7)	976 (66.4)
**Was in a physical fight in the last 12 months *^,+^** No times Any times	4870 (82.3) 1044 (17.7)	3215 (71.1) 1306 (28.9)	1626 (20.5) 6289 (79.5)	1293 (25.0) 3882 (75.0)
**Texted and drove in the past 30 days (among those who drove) *^,+^**				
No days	1193 (67.4)	1061 (50.7)	2009 (59.6)	1206 (63.2)
Any days	576 (32.6)	1029 (49.3)	1360 (40.3)	701 (36.8)
**Risk Factors**				
**Initiation of alcohol consumption ^+^**				
Never	3553 (60.3)	1680 (37.4)	4257 (53.8)	2507 (48.3)
13 or younger	1380 (23.4)	1678 (37.3)	2078 (26.3)	1696 (32.7)
14 or older	955 (16.2)	1138 (25.3)	1555 (19.7)	935 (18.0)
**Protective Factors**				
**Eat dinner with family *^,+^** Usually	3968 (68.1)	2551 (57.8)	5211 (67.0)	2739 (54.0)
Not usually	1859 (31.9)	1864 (42.2)	2565 (33.0)	2332 (46.0)
**Getting along with parents *** Completely true Not completely true	2007 (36.5) 3490 (63.5)	1158 (28.0) 2982 (72.0)	2455 (33.2) 4939 (66.8)	1522 (32.7) 3130 (67.3)
**Have adult support to turn to if feeling sad/hopeless *^,+^** Yes No Not sure	3362 (72.5) 622 (13.4) 654 (14.1)	2188 (63.0) 679 (19.6) 605 (17.4)	4307 (70.5) 897 (14.7) 908 (14.9)	2407 (60.1) 852 (21.3) 744 (18.6)
**Feel safe at school ***^,+^ Yes No	5344 (90.3) 576 (9.7)	3816 (85.6) 640 (14.4)	7134 (90.1) 786 (9.9)	4358 (84.1) 821 (15.9)
**Think about consequences before making a decision *^,+^** Strongly agree/agree Disagree/strongly disagree	4559 (87.4) 655 (12.6)	3228 (81.7) 723 (18.3)	6097 (86.2) 976 (13.8)	3568 (81.9) 788 (18.1)

** Column percentages. * *p* < 0.01 significant difference among life jacket wearers and non-wearers. ^+^
*p* < 0.01 significant difference among formal swimming lessons and no formal swimming lessons.

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
