# Peer review of "Adolescent Water Safety Behaviors, Skills, Training and Their Association with Risk-Taking Behaviors and Risk and Protective Factors"

_children, 2020, doi:10.3390/children7120301_

Round 1
Reviewer 1 Report
Thank you for the opportunity to review manuscript ‘Adolescent Water Safety Behaviors, Skills, Training and Their Association with Risk-taking Behaviors and Risk and Protective Factors’. This manuscript reports the findings and correlation between swimming lessons, risk taking behaviours and sociodemographic characteristics of individuals of publicly available data set in the USA. Overall, this manuscript is generally well written and addresses a gap in the drowning prevention literature using a novel approach. I have made some overall comments, with a few specific comments relating to page and line numbers below. I hope the comments are found helpful to the authors for improving their manuscript.
Overall comments
There are a number of minor grammatical issues in the manuscript. I tried to make reference to specific points below, however, suggest the authors complete an additional proof-read of the document before re-submitting. There are several vary lengthy sentences, which loose meaning when reading. Not all sentences were highlighted below in comments.
In the abstract, the reference to adolescent water safety training (swimming lessons) is used. For consistency, it is recommended using only one term ‘swimming lessons. As swimming lessons is not defined in the paper, it is not understood if the authors refer to ‘swimming lessons’ in the traditional form (i.e. stroke formation). Water safety training has an ambiguous meaning which could include swimming, lifejacket wearing, throw ropes, survival water techniques, lifeguard rescues etc. The need for consistent language within the drowning prevention community has been highlighted by a number of reviews.
The use of the terms Black and White. Are these appropriate for an international audience or should terms such as Caucasian and African American be more culturally appropriate.
Is there any studies relating the same demographics and characteristics (low socioeconomic, language etc) to high levels of injury in the same age cohort. Could this be presented more strongly in the discussion, the relationship between drowning and other injuries in the cohort, and highlight any major difference is any present.
Stronger relationship needs to be drawn between current preventable activities for other injuries in the age cohort and relate back to possible drowning prevention activities for the same cohort.
In the manuscript reviewed, no references in text could be located. It is suggested that this is amended.
Specific comments
Abstract
Line 15 – adolescent water safety training (swimming lessons), please use one term for consistency
Line 18 – To make the data more comparable to other studies an age qualification would be useful (an age rang in years for the children surveyed)
Lie 21 – is White the correct term to use? Caucasian?
Line 24 – swimming lessons is not defined in text. It is suggested that swimming lessons is defined in text to represent its meaning for this study
Introduction
Line 34 – requires a reference
Line 36 – requires a reference
Line 37 – requires a reference (please reference all statistics or information from sources). No further comments to referencing will be made below
Line 55 – define formal swimming lessons. What is a formal swimming lesson in relation to this study?
Materials and Methods
Line 66 – it would be useful for a qualification of age to be presented in this study i.e. age of students ranged from 12 to 18 years (median age and SD). This will help for comparable analysis in the future between studies.
Line 88 – were students required to read the survey/complete survey individually – what if they were not of English speaking backgrounds?
Line 129 – exempt from what? Exempt from ethics? Please provide further information.
Results
Line 140 – identified as white or Caucasian? Please for consistency refer to one term
Line 167 – Black or African and American? Again for consistency refer to one term. It is it culturally appropriate to use Black and White or is Caucasian and African American etc preferred?
Line 198 – some stood out. How many is some? And is it identified not stood out?
Line 212 – Importantly? Why is it important. Please delete. Simply it is demonstrated.
Line 127 – make sure all reference of previous work or statistics is added
Line 221 – what is the marked difference or the statistical difference between this age group and the drowning rate seen in the USA for context in the discussion. Provide reference
Line 227- reference
Line 228 – what are the presented dangers for overestimated swimming ability in relation to this age cohort?
Line 252 – very lengthy sentence, loss of meaning/heard to read
Line 292 – “several”, please quantify
Lien 291 – limitation of reading and writing to complete survey? Bias shown in results due to the cofound factors between this and swimming ability/life jacket wearing
Conclusion
Line 320 – Usefulness predicted, unsure of meaning of this sentence, and what it relates to
Reviewer 2 Report
This is an important and well done study identifying some of the behaviors, skills, training and risk taking behaviors and risk and protective factors in adolescent water safety. Boy, that's a mouthful. Maybe the title is biting off more than can be chewed. Perhaps breaking it down to what are the most important purposes first (is it behaviors, skills, training, risk taking behaviors risk and protective factors?), then highlighting those in the results.
If the study was partially in-kind supported by the Harborview Injury Prevention Research Center (Funded by CDC), then the acknowledgements should read "in-kind support from the Harborview Injury Prevention and Research Center funded by the Injury Center at the Centers for Disease Control and Prevention Grant # ________ ".
Please clarify (line 51-52) if there were question overlap between the CDC YRBS, and HYS and if you looked at those overlaps for comparison purposes. for the data in grades 8-12. Was the HYS a statewide version of the YRBS?
Lines 34-53. There are no references listed, attached to the various statements on burden (34-40) or risk factors (41-52). Add references to those statements (e.g. 43-44).
You mention line 77 that 8th grade is included in your sample, yet line81 you say the HYS surveys high school students (not 8th grade...is 8th grade in Wash State high school?)
Line 102 (Jessor), needs a reference (ref 11?)
line 105 Do you have any reliability and validity to report?
lines 106-109. Please note somewhere that those risk and protective were not related to aquatic behavioral factors, but to general lifestyle factors.
line 119-120 Do you have a references for these statements? Please provide
line 132-169 are any of these results in any of the tables besides Table 1? There are no P values in the table. There is no description in the test about which statistical method you used to derive the P less than 0.01. Similar for line 173-187.
Line 210-213 should be one paragraph
Line 216-217 needs a reference so does line 224-230. & line 235-236 and much of the cited facts in lines 237-248
line 269-270 should reference Jessor (ref 11?)
Reference section
Ref 1 is cited incorrectly. It should be as follows: Centers for Disease Control and Prevention. National Center for Injury Prevention and Control. Web-Based injury Statistics Query and Reporting System (WISQARS). [on-line] Accessed 1 July 2020. Available from URL www.cdc.gov/injury/wisqars
ref 12 needs more information is 10,12,13 the same?
